# The Emotion Regulation Mechanism in Neurotic Individuals: The Potential Role of Mindfulness and Cognitive Bias

**DOI:** 10.3390/ijerph20020896

**Published:** 2023-01-04

**Authors:** Ling Chen, Xiqin Liu, Xiangrun Weng, Mingzhu Huang, Yuhan Weng, Haoran Zeng, Yifan Li, Danna Zheng, Caiqi Chen

**Affiliations:** 1School of Psychology, South China Normal University, Guangzhou 510631, China; 2Key Laboratory of Brain, Cognition and Education Sciences (South China Normal University), Ministry of Education, Guangzhou 510631, China; 3Center for Studies of Psychological Application, South China Normal University, Guangzhou 510631, China; 4Guangdong Key Laboratory of Mental Health and Cognitive Science, South China Normal University, Guangzhou 510631, China; 5School of Foreign Languages, South China University of Technology, Guangzhou 510641, China

**Keywords:** neuroticism, emotion regulation, mindfulness, cognitive bias

## Abstract

Neuroticism is a personality trait that impacts daily life and raises the risk of mental problems and physical illnesses. To understand the emotion regulation mechanism of neurotic individuals, we developed two complementary studies to examine the effects of mindfulness and negative cognitive bias. In Study 1, four scales (EPQ-RSC, FFMQ, CERQ, NCPBQ) were used for assessment. Correlation analysis and structural comparison showed that: (1) the level of neuroticism was positively correlated with negative emotion regulation; (2) negative cognitive bias mediated the relationship between neuroticism and emotion regulation; (3) mindfulness and negative cognitive bias mediated the relationship in a chain. Study 1 showed that cognitive bias may play a key role in the emotion regulation mechanism. Study 2 further explored the cognitive bias of neurotic individuals using three behavioral experiments. A mixed-design ANOVA indicated that individuals with high neuroticism levels exhibited negative attention, memory, and interpretation biases. Our findings extend previous research on emotion regulation problems of neurotic individuals and broaden the field to personality-based emotion disorders. In particular, a theoretical rationale is provided for the application of cognitive behavioral therapy, such as mindfulness-based cognitive therapy (MBCT), to the emotion regulation of neurotic individuals.

## 1. Introduction

Neuroticism is a personality trait associated with emotional instability. Because of its importance to the lives of individuals and to public health as a whole, it has recently received increased attention. Neuroticism is considered a danger indicator and a general risk factor for the onset and development of mental disorders [1,2,3]. Previous research has shown that neuroticism is a strong predictor of suicidal thoughts [3]. Individuals with high neuroticism are prone to experience more negative emotions, such as anxiety or worry [4], which may be due to negative emotion regulation. However, little is known about the underlying regulatory mechanisms in neurotic individuals. Therefore, it is necessary to examine the characteristics and factors of their emotion regulation. 

A negative correlation exists between neuroticism and emotion control [5,6]. Emotion regulation refers to the internal and external processes by which a person monitors, evaluates, and corrects emotional responses in order to achieve goals [7]. Previous studies demonstrated that neuroticism is inversely related to an individual’s tendency to use cognitive reappraisal strategies, a type of emotion regulation. Individuals with low neuroticism levels were more effective at regulating negative emotions due to their tendency to use cognitive reappraisal strategies. Therefore, they experienced fewer negative emotions. In contrast, individuals with high neuroticism had poor emotion regulation and poor emotional state [8]. On this basis, our first proposed hypothesis is as follows:

**H1.** *The level of neuroticism is positively correlated with negative emotion regulation*.

According to the orienting attention/action readiness (OAAR) framework, the interplay between action readiness and orienting attention determines the outcomes of emotional regulation [9]. On the one hand, action readiness of a specific regulation strategy increases the probability of that strategy being used in similar emotional circumstances [10]. Cognitive bias, a preference for assessing, processing, selecting, and remembering precise information [11,12,13], increases the likelihood of performing the strategy. Cognitive bias research has shed light on the relationship between neuroticism and emotional regulation. According to Disner’s neurocognitive hypothesis, individuals with negative traits form negative cognitive schemas triggered by environmental factors, manifesting as negative attention, interpretation, memory, and rumination biases [14]. Other researchers argue that highly neurotic individuals have negative emotional responses influenced by various cognitive biases, and that negative cognitive biases lead to more negative emotional experiences [15,16,17]. On this basis, Hypothesis 2 was established: 

**H2.** *Negative cognitive bias plays a mediating role between neuroticism and negative emotion regulation*.

Focusing attention causes individuals to perform various top-down actions faster, increasing the frequency of using specific emotion regulation [9]. Previous studies showed that people trained in meditation exhibit high levels of mindfulness and increased orienting attention [18]. Mindfulness refers to purposeful, non-judgmental attention in the present moment [19]. Dispositional mindfulness (or trait mindfulness) is similar to character strength in positive psychology [20] and refers to an individual’s ability to remain aware and focused on the experience of the moment [21]. Numerous studies indicate a significant negative correlation between neuroticism and mindfulness [22,23,24,25]. Individuals with high neuroticism are fearful of potential future threats and engrossed in previous negative emotional experiences [26,27]. They have more difficulty focusing on the present moment [28,29]. Based on the above analysis, Hypothesis 3 is proposed: 

**H3.** *Mindfulness plays a mediating role between neuroticism and negative emotion regulation*.

The OAAR framework proposes that action readiness (e.g., cognitive bias) and orienting attention (e.g., mindfulness) can work together to influence emotion regulation [9]. According to cognitive behavioral theory, one’s irrational cognition of events, especially the cognitive bias towards negative information, such as attention and interpretation bias, is the root cause of negative emotions [30,31,32]. Mindfulness facilitates the connection between past and current emotional experiences, guiding future emotional understanding and regulatory adaptation [33]. Consequently, people with a high level of mindfulness are more conscious of their current emotions and perceptions [34]. They have reduced levels of cognitive bias and a more objective understanding of their emotional responses to life events [35]. Based on these findings, we suggest Hypothesis 4:

**H4.** *Mindfulness and negative cognitive bias play a chain mediating role between neuroticism and negative emotion regulation*.

Although previous studies on the emotion regulation of neurotic individuals provide essential findings, the mechanisms underlying the regulation are still not well known. Most of these studies have not focused on the cognitive aspects of neuroticism and, methodologically, they only used scales for assessment. This study extends previous research with two additional studies using scales and experiments. Study 1 investigated the association between neuroticism and negative emotion regulation, and examined the role of mindfulness and negative cognitive bias. Based on Study 1, Study 2 used three experiments to delve into the cognitive bias performance of neurotic individuals. Combining the two methods helps understand the underlying mechanisms with a focus on the role of mindfulness and negative cognitive bias. This has clinical implications for cognitive behavioral therapy. We also offer suggestions for investigating the relationship between neuroticism and other psychiatric disorders, and provide a reference for identifying and intervening in the case of psychological crises. The model shown in Figure 1 combines the four possibilities.

## 2. Study 1: Scale Study of the Emotion Regulation Mechanisms in Neurotic Individuals 

### 2.1. Method

#### 2.1.1. Participants

Respondents were randomly recruited from Chinese university students. The sample consisted of 260 students (mean age = 19.20 years, SD = 1.14; 198 females). 

#### 2.1.2. Procedure

Data were collected via online questionnaires.

#### 2.1.3. Measures

Neuroticism. Neuroticism was measured using the Eysenck Personality Questionnaire-Revised Short Scale for Chinese (EPQ-RSC). The questionnaire consists of 48 items, categorized into four dimensions (extraversion, neuroticism, psychoticism, lie), and has high reliability and validity [36]. Respondents were required to provide yes-or-no responses. The formula for calculating the standard score was T = 50 + 10 ((subject’s original score − the mean of norm)/standard deviation of norm).

Mindfulness. The Five Facet Mindfulness Questionnaire (FFMQ), created by Baer [37] and amended by Deng [38], was used to assess mindfulness. There are 39 items on the scale, which can be divided into five dimensions (observe, describe, act with awareness, nonjudge, and nonreact). This is a 5-point Likert scale, and each item contains five response options, ranging from 1 (complete nonconformity) to 5 (complete conformity). Higher scores indicate greater mindfulness. In this study, Cronbach’s alpha for the scale was 0.813.

Emotion regulation. Individual emotion regulation strategies were evaluated using the Cognitive Emotion Regulation Questionnaire (CERQ) [39]. This questionnaire consists of 32 items that measure eight aspects of emotion regulation: self-blame, other-blame, rumination, catastrophizing, positive refocusing, positive reappraisal, putting into perspective, and acceptance. Among them, positive refocusing and positive reappraisal concern positive responses, while the others concern negative responses. In this study, Cronbach’s alpha scale was 0.866.

Negative cognitive bias. The Negative Cognitive Processing Bias Questionnaire (NCPBQ), compiled by Zhang [40], consists of 24 items categorized into four dimensions: negative attention bias, negative memory bias, negative interpretation bias, and negative meditation bias. This is a 4-point Likert scale (1 = not at all satisfy, 4 = fully satisfy). In this investigation, Cronbach’s alpha scale was 0.906.

#### 2.1.4. Analysis

The statistical analysis was performed using software such as MS Excel 2019, IBM SPSS 25.0, and Mplus 8.0 [41].

### 2.2. Results

#### 2.2.1. Common Method Bias

Harman’s single-factor test was used as a statistical instrument to evaluate common method bias. In an unrotated principal component analysis of all measured items using Harman’s univariate test, the first common factor accounted for 13.3349% of the total variance, substantially less than 40% (cutoff value). Thus, common method bias did not significantly affect our study.

#### 2.2.2. Correlation Analysis

Correlations were analyzed between four variables: neuroticism, mindfulness, negative cognitive bias, and negative cognitive bias. Table 1 presents the results. Neuroticism was negatively correlated with mindfulness (r = −0.364, *p* < 0.001) and positively associated with negative cognitive bias (r = 0.734, *p* < 0.001) and negative emotion regulation (r = 0.551, *p* < 0.001). Mindfulness was negatively correlated with negative cognitive bias (r = −0.388, *p* < 0.001) and negative emotion regulation (r = −0.290, *p* < 0.001). Negative emotion regulation was positively correlated with negative cognitive bias (r = 0.597, *p* < 0.001).

#### 2.2.3. Test of the Chain Mediation Model

The structural equation modeling method [42] was used to test the chain mediation effect. The measurement model consisted of three dominant variables (neuroticism, mindfulness, and negative emotion regulation) and one latent variable: negative cognitive bias. The latter was derived from four dominant variables (negative attention, memory, interpretation, and rumination bias), and gender was included as a control variable.

First, the measurement model was tested, and all fit indicators of the model met the requirement for a satisfactory fit standard: χ^2^/df = 1.977, CFI = 0.982, TLI = 0.963, RMSEA = 0.061, and SRMR = 0.027. The paths in this model were then analyzed, and the results are shown in the path model in Figure 2.

The significance of mediation effects was tested using the Bias-Corrected Bootstrap procedure, and 5000 samples were selected for iterative analysis. As can be observed in Table 2, the mindfulness mediation impact was not significant; negative cognitive bias accounted for 90.42% of the total effect, and the chain mediation effect of positive and negative cognitive bias was significant, accounting for 7.78% of the total effect.

### 2.3. Discussion

Study 1 used four scales to examine the emotion regulation mechanism in neurotic individuals. Consistent with the findings of prior research [43,44], the results demonstrated that the individual’s negative emotion control was proportional to their neuroticism level. Meanwhile, the mediation effect of a negative cognitive bias was significant, and both mindfulness and negative cognitive bias acted as chain mediators between neuroticism and negative emotion regulation. The mediation effect of mindfulness was not significant, indicating that cognitive bias may play a vital role in the emotion regulation mechanism of neurotic individuals. Study 2 used experimental methods to investigate the cognitive bias of neurotic persons further.

## 3. Study 2: The Experimental Study of Cognitive Bias in Neurotic Individuals

### 3.1. Method

#### 3.1.1. Participants

We asked 145 undergraduates from a Chinese university to complete the Eysenck Personality Questionnaire. According to the experimental requirements, three groups of individuals were selected: a high neuroticism group (standard score T > 61.5), a low neuroticism group (standard score T < 38.5), and a medium control group (standard score T between 38.5 and 61.5). As each of the three groups consisted of 30 individuals, the final sample comprised 90 individuals (mean age = 19.52 years; standard deviation = 1.40; 61 females).

A one-way ANOVA was performed to test the validity of the subjects’ grouping. They demonstrated highly significant differences in the standard scores of the three groups on the neuroticism dimension, F(2, 87) = 281.104, *p* = 0.000. The subject grouping was genuine and met the experimental requirements. Results are shown in Table 3.

#### 3.1.2. Design

The attention bias experiment was a 3 (group: high neuroticism, low neuroticism, and medium control) × 5 (experiment condition: positively consistent, positively inconsistent, negatively consistent, negatively inconsistent, and neutral) mixed design, with the group serving as the between-subject factor, and the experiment condition as the within-subject factor. Again, the dependent variable was the time required to identify the emotional image.

The memory bias experiment consisted of a 3 (group: high neuroticism, low neuroticism, and medium control) × 3 (emotional picture type: positive, negative, and neutral) mixed design, with the group serving as the between-subject factor, and the emotional picture type serving as the within-subject factor. The dependent variable was the memory retention of emotional pictures.

The interpretation bias experiment utilized a 3 (group: high neuroticism, low neuroticism, and medium control) × 2 (interpretation type: positive and negative) mixed design, with the group serving as the between-subject factor, and interpretation type as the within-subject factor. The dependent variables were endorsement rate and reaction time for word recognition.

#### 3.1.3. Materials

The emotional pictures used in the experiment on attention bias were chosen from Wang’s compilation of the Chinese Facial Affective Picture System (CFAPS) [45]. Randomly selected from CFAPS were images of cheerful faces (positive stimulus), sad faces (negative stimulus), and calm faces (neutral stimulus). Randomly pairing the emotional images resulted in 24 positive–neutral pairs, 24 negative–neutral pairs, and 12 neutral–neutral pairs.

Similarly, 90 emotional pictures were randomly selected from CFAPS for the memory bias experiment, with 30 in each type (positive, negative, and neutral emotional faces). A total of 45 pictures were used in the learning phase (i.e., the old pictures) and the other 45 in the memory recognition test phase (i.e., the new pictures).

The stimulus material for the interpretation bias experiment was partly modified based on the text material used in Dearing’s study [46]. A total of 30 ambiguous situational sentences were supplemented with multiple aspects of life events, and each situation sentence had two explanatory words. There are 30 benign words and 30 negative words. The benign words include both neutral and positive words.

#### 3.1.4. Procedure

Attention bias was tested using a dot-probe paradigm (see Figure 3). First, a fixation cross was presented in the center of the screen with a duration of 600 ms, to which participants had to stare. Next, a blank screen of 14 ms was displayed. Then two faces appeared symmetrically (one on the left side of the screen and one on the right) for 500 ms. A blank screen of 14 ms was then displayed. Next, a probe stimulus “○” appeared at the location of one of the faces for 1500 ms. The participant’s task was to quickly determine whether the stimulus “○” was displayed on the left or right by hitting the “F” or “J” key. After the participants had responded (or if they did not respond within 2000 ms), the screen would remain blank for 1000 ms until the next trial began. Prior to the formal experiment, there were practice tryouts of 100 trials. It was a consistent trial if the probe stimulus occurred in the same position as an emotional face. Otherwise, the trial was inconsistent. As the emotional faces fell into three categories (positive, negative, and neutral), a trial was considered positively consistent if it had a cheerful face. Thus, there were five trials (or experimental conditions): positive consistent, positive inconsistent, negative consistent, negative inconsistent, and neutral conditions. The order in which stimuli were delivered was arbitrary.

A recognition memory paradigm was used to examine memory bias (as shown in Figure 4). The experiments consisted of three stages: learning, interference, and recognition. In the learning stage, 45 trials were performed. First, a fixation cross was presented in the center of the screen for 500 ms. Next, an image of a random face was displayed for 4000 ms, and participants were required to judge their familiarity with the picture (three options while consciously memorizing the image: 1. familiar with the picture, or have seen a similar picture; 2. unfamiliar with the picture, or have not seen a similar picture; 3. unsure). During the interference stage, participants performed 15 uncomplicated arithmetic calculations to mask the impact of subsequent learning in the recognition stage. There were 90 trials in the recognition stage. First, a fixation cross was presented in the center of the screen for 500 ms. Next, an image of a random face was presented for 2000 ms, and participants were asked to recognize the type of image (two choices: 1. old image; 2. new image).

A word–sentence association paradigm (WSAP) was used to examine interpretation bias (as shown in Figure 5). First, a fixation cross was displayed for 500 milliseconds. This alerted participants that a trial was about to begin by drawing their attention to the screen. Second, a term conveying a benign association (such as “funny”) or a threat association (such as “embarrassing”) was displayed for 500 milliseconds. Third, for 2000 milliseconds, one ambiguous sentence (such as “People laugh after saying something”) would be displayed. Participants were instructed to press the “F” key if they believed the term and the sentence were related after the sentence disappeared. Otherwise, they had to press the “J” key. Before the formal experiment with 30 trials, there were five practice trials. The stimuli were delivered in a random order for each participant.

#### 3.1.5. Analysis

IBM SPSS 25.0 and MS Excel 2019 were used for statistical analysis. The experimental data were evaluated based on response time and accuracy. The outliers of response times (mean ± 3 standard deviations) were eliminated.

### 3.2. Results

#### 3.2.1. Attention Bias

The descriptive statistics of the probing stimulus responses of the three participant groups under the five experimental conditions were collected. Results are shown in Table 4.

This attention bias index was calculated as follows: Positive attention bias = RT of positive inconsistent—RT of positive consistent. Negative attention bias = RT of negative inconsistent—RT of negative consistent. Descriptive statistics of the positive and negative attention biases are shown in Table 5.

These attention bias index scores were subjected to a 3 × 2 mixed-design ANOVA. The between-group factor was the Neuroticism Group (three levels: high neuroticism, low neuroticism, and medium control). The within-group factor was Attention Bias Type (two levels: positive and negative). The main effect of the Attention Bias Type was not significant [F(1, 87) = 0.243, *p* = 0.623]. The main effect of group was also not significant [F(2, 87) = 0.983, *p* = 0.378]. However, the interaction between the two factors was significant [F(2, 87) = 8.021, *p* = 0.001, *η*^2^ = 0.156] (see Figure 6).

A further simple effect analysis revealed that the mean response time for recognizing negative emotional faces in the high neuroticism group was significantly longer than for cheerful faces [F(1, 87) = 8.128, *p* = 0.005, *η*^2^ = 0.085]. The low neuroticism group’s average recognition time for cheerful emotional faces was substantially longer than for negative emotional faces [F(1, 87) = 7.05, *p* = 0.009, *η*^2^ = 0.075]. In addition, there were significant variations in the response times of the different groups for recognizing cheerful emotional faces [F(2, 87) = 7.408, *p* = 0.001, *η*^2^ = 0.146]. Similarly, there were significant differences between the high and low neuroticism groups and between the high and medium neuroticism groups. In addition, the findings revealed an important group difference in response times for recognizing negative emotional faces [F(2, 87) = 3.259, *p* = 0.043, *η*^2^ = 0.070], and a meaningful difference existed between the high and low neuroticism groups.

#### 3.2.2. Memory Bias

The retention of positive, negative, and neutral pictures was measured. Memory bias index = (number of correct recognitions—number of incorrect recognitions)/(number of study pictures + number of new pictures) × 100%. Table 6 provides the results.

These memory bias index scores were subjected to a 3 × 3 mixed-design ANOVA. The between-group factor was the Neuroticism Group (high neuroticism, low neuroticism, and medium control), and the within-group factor was the Emotional Picture Type (positive, negative, and neutral). Data analysis revealed a significant main effect of the emotional picture type [F(2, 174) = 16.898, *p* < 0.001, *η*^2^ = 0.163]. Subjects remembered negative emotional pictures better. The main effect of group was not significant [F(2, 87) = 0.960, *p* = 0.387]. The interaction between the two factors was also not significant [F(4, 174) = 0.915, *p* = 0.455].

#### 3.2.3. Interpretation Bias

The WSAP resulted in four reaction time variables, response latencies for (1) endorse threat, (2) reject threat, (3) endorse benign, and (4) reject benign. We then obtained two primary interpretation indices: (1) the percentage of endorsed benign interpretations and (2) the percentage of endorsed interpretations with threats. 

Descriptive statistics were obtained from the responses of the three groups of participants, and the results are shown in Table 7.

The percentage of endorsed benign and threat interpretations was used as the interpretation bias index. Descriptive statistics are shown in Table 8.

These interpretation bias index scores were subjected to a 3 × 2 mixed-design ANOVA. The between-group factor was the Neuroticism Group (high neuroticism, low neuroticism, and medium control), and the within-group factor was the Interpretation Bias Type (positive and negative). Data analysis revealed a significant main effect of the interpretation bias type [F(1, 87) = 4.343, *p* = 0.040, *η*^2^ = 0.048]. In contrast, the main effect of group was not significant [F(2, 87) = 1.848, *p* = 0.164]. However, the interaction between the two factors was significant [F(2, 87) = 4.142, *p* = 0.019, *η*^2^ = 0.087] (see Figure 7).

A further simple effects analysis showed that individuals with high neuroticism had a significantly more negative than positive interpretation bias [F(1, 87) = 12.132, *p* = 0.001, *η*^2^ = 0.122]. In addition, there were substantial differences in the negative interpretation bias between different groups [F(2, 87) = 5.183, *p* = 0.007, *η*^2^ = 0.106]. This indicated notable differences between the groups with high and low neuroticism, and those with high and moderate neuroticism.

### 3.3. Discussion

The attentional bias of neurotic individuals was examined using a dot-probe paradigm. The experimental findings revealed that the primary effect of the emotional face type was not statistically significant. Neither was the primary effect of the level of neuroticism. Consistent with earlier findings [47,48], individuals with high neuroticism tended to have a negative attentional bias, whereas individuals with low neuroticism had a positive attentional bias.

We used the recognition memory paradigm to examine the memory bias of neurotic individuals. The experimental results indicated that the main effect of the emotional face type was significant. Regardless of whether their neuroticism level was high, low, or medium, individuals had a memory bias for negative images. The main effect of the neuroticism group was not significant. The interaction between the emotional face type and the neuroticism group was also not significant. Previous research revealed that those with high neuroticism displayed a negative memory bias, whereas those with low neuroticism exhibited a positive memory bias. However, other research suggests that all individuals recognize negative emotional faces better than positive or neutral ones. According to the negative emotion priority theory, and the threatening or negative emotion primacy theory, threatening or negative emotion information is processed in preference to positive or neutral emotion information, which is a human preference to avoid harm in response to the environment, allowing individuals to better adapt to changes [49,50].

The word sentence association paradigm (WSAP) was used to investigate the interpretation bias of neurotic individuals. The experimental results show that the endorsement rate was significant. The negative endorsement rate was significantly higher than the benign reinforcement rate. The interaction between interpretation bias and neuroticism was also significant. Individuals with high neuroticism had a negative interpretation bias, consistent with the results of previous studies. According to these [51], individuals with high social anxiety are more likely to exhibit a negative interpretation bias than those with low social anxiety. Depressed individuals have a considerable negative bias in the immediate interpretation stage of processing ambiguous information [52].

## 4. General Discussion

This study examined the mechanisms of emotion regulation in neurotic individuals through a combination of questionnaire research and experimental investigation, focusing on the role of mindfulness and cognitive bias. 

### 4.1. Emotion Regulation Mechanisms of Neurotic Individuals

First, consistent with Hypothesis 1, Study 1 showed a positive correlation between neuroticism and negative emotion regulation. The result showed that the higher the level of neuroticism, the more negative emotion regulation. The results suggest that neuroticism can be seen as a potential developmental factor with differences in the tendency to use adaptive or maladaptive emotion regulation strategies [53]. The Five-Factor model of personality is an important theory [54]. Of these five factors, neuroticism is associated with emotional instability and negative outcomes, and even predicts emotional disorders such as depression and anxiety [1,55,56].

Furthermore, consistent with Hypothesis 2, we found that a cognitive bias mediated the relationship between neuroticism and emotion regulation. According to the Emotion Regulation Process Model [57], individuals with low neuroticism enjoy more emotional stability. They are more likely to adopt rapidly effective ways of evaluating negative emotion-eliciting situations, including a positive cognitive bias toward current events, reducing their own experience and response to negative emotions. Individuals with a high level of neuroticism are oversensitive and unresponsive to emotional stimuli and adopt maladaptive regulation strategies. 

However, contrary to Hypothesis 3, mindfulness was not a mediator in the relationship between neuroticism and emotion regulation. One possible explanation is that cognitive bias plays a crucial role in emotion regulation. In other words, the effect of mindfulness is nested under the influence of cognitive bias. Since dispositional mindfulness has adaptive psychological effects, it can improve the individual’s cognitive function, increase their understanding of emotions and emotional cues, and thus improve their emotional regulation [58,59]. According to the mindfulness-to-meaning theory, the decentralized mechanism of mindfulness can reduce automated cognitive responses, improve cognitive flexibility, and ultimately stimulate the individual’s positive emotions [60].

Finally, Study 1 examined whether mindfulness and cognitive bias play a chain mediating role between neuroticism and emotion regulation, and the results support Hypothesis 4. Mindfulness, characterized by awareness and acceptance of ongoing emotional experiences [61], can serve as a protective factor against the negative processes associated with neuroticism. Previous studies have suggested that mindfulness is correlated with neuroticism and negative emotions [22]. According to the orienting attention/action readiness (OAAR) framework, the interaction between action readiness and orienting attention determines emotional regulation outcomes [13]. Study 1 provided empirical support for the OAAR framework by using mindfulness and cognitive bias as mediating variables. 

These results suggest that we can improve the emotion regulation of neuroticism by increasing their level of mindfulness, and then improving the cognitive bias. The reperceiving model suggests that mindfulness can regulate modes of perceiving, which allows people to focus on the moment-to-moment experience [33]. Mindfulness reduces the distortion of the emotional valence caused by stimuli, making cognition more neutral and objective, increasing behavioral flexibility and resulting in a more positive cognitive bias. Therefore, future studies can make more efforts to investigate whether the intervention of mindfulness can reduce the negative emotion regulation of neuroticism. Furthermore, mindfulness training can be used in preventing the increase in negative emotion in neurotic individuals by improving their mental flexibility in response to the environment. The application of mindfulness training may also be generalized to non-neurotic population.

### 4.2. Cognitive Bias of Neurotic Individuals

Study 1 found that mindfulness and negative cognitive bias played a chain mediating role between neuroticism and negative emotion regulation. However, the mediating effect of mindfulness was not significant, suggesting that cognitive bias was a critical factor in the emotion regulation mechanism of neurotic individuals.

Based on Study 1, Study 2 used three experiments to examine the cognitive bias performance of neurotic individuals. The results indicated that people with high neuroticism have attentional, memory, and interpretation biases in response to negative emotional stimuli. Individuals with high neuroticism focus on the negative aspects of things and always process information negatively, which means that they are more susceptible to negative information stimuli and have negative cognitive processing biases, such as negative attention, memory, and interpretation biases.

The following are the outcomes of Study 2’s three experiments: First, in the attention bias experiment, and under different experimental paradigms, neurotic individuals exhibited a strong attention bias toward negative stimuli, as demonstrated by a greater allocation of attention to negative stimuli and difficulties disengaging from negative stimuli [62]. Second, in the memory bias experiment, individuals with high neuroticism can have a negative memory bias and recall or identify specific prior experiences that are consistent with their mental state, i.e., to choose emotional information in line with their personality traits [7]. Last, in the interpretation bias experiment, persons with high neuroticism interpreted events with negatively and developed negative schemas for stimuli, resulting in a negative interpretation bias.

### 4.3. Limitations and Future Possibilities

This study has some limitations. First, it did not cover all aspects of cognitive bias in neurotic individuals. According to the neurocognitive theory of negative cognitive bias, four aspects should be included: negative attentional bias, interpretation bias, memory bias, and rumination bias [14]. This study omitted the final component (rumination bias). Future research could evaluate whether there are interactions among these four types of bias. Second, trait mindfulness was the only variable used in this study to quantify mindfulness. Future studies could examine the effect of mindfulness training on improving emotion regulation mechanisms. This may provide more empirical evidence for the application of cognitive behavioral therapy to neuroticism treatment. Third, this study only explored the emotion regulation of neurotic individuals through scales and behavioral experiments. Future research could integrate multiple methods, such as behavioral experiments and brain imaging technology (EEG, ERP, fMRI, etc.), to investigate the emotion regulation mechanisms of neurotic individuals, such as the identification of the cortical network associated with brain mechanisms underlying emotion regulation [63]. Fourth, the present study’s cross-sectional design and mediation analysis were unable to establish causality [64]. Future research could design longitudinal studies to specify the directionality of the relationship between variables. Fifth, since some of the previous studies showed that people suffering from depression had higher levels of neuroticism [65], future studies should consider and control for the role of depression. In addition, the current study only recruited university students, which might limit the generalizability of the findings. Therefore, other groups of people should be examined in future studies.

### 4.4. Implications

Our study has various theoretical and practical consequences, despite its limitations. From a theoretical perspective, the findings extend the previous research on emotion regulation problems of neurotic individuals, broaden the scope of studies on personality-based emotion disorders, and provide evidence supporting the OAAR paradigm. From a practical standpoint, our study focused on the cognitive aspects of neuroticism, which may offer valuable insights into how to alleviate the emotion regulation problems of neurotic individuals. In particular, it provides empirical evidence and theoretical support for the application of cognitive behavioral therapy, such as mindfulness-based cognitive therapy (MBCT). It should help highly neurotic individuals to recognize the occurrence of potentially unhelpful processes, thereby improving and enhancing emotion regulation [66] and further improving their mental health.

## 5. Conclusions

This study combined a questionnaire survey and three experiments to investigate the emotion regulation mechanism of neurotic individuals. The main findings are: (1) The level of neuroticism positively correlated with negative emotion regulation; (2) Negative cognitive bias mediated the relationship between neuroticism and negative emotion regulation; (3) Mindfulness and negative cognitive bias mediated the relationship in a chain; (4) Individuals with high neuroticism tended to show negative attention, memory, and interpretation bias. This study enhances our understanding of the emotion regulation mechanism of neurotic individuals and offers a fresh perspective for future research into the emotional and cognitive factors associated with neuroticism. It may have theoretical, practical, and methodological implications for researchers and professionals in the field of personality-related emotion disorders.

## Figures and Tables

**Figure 1 ijerph-20-00896-f001:**
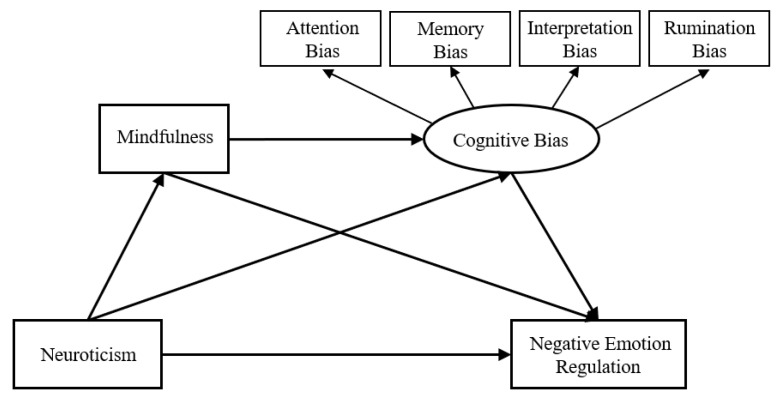
Hypothesis model.

**Figure 2 ijerph-20-00896-f002:**
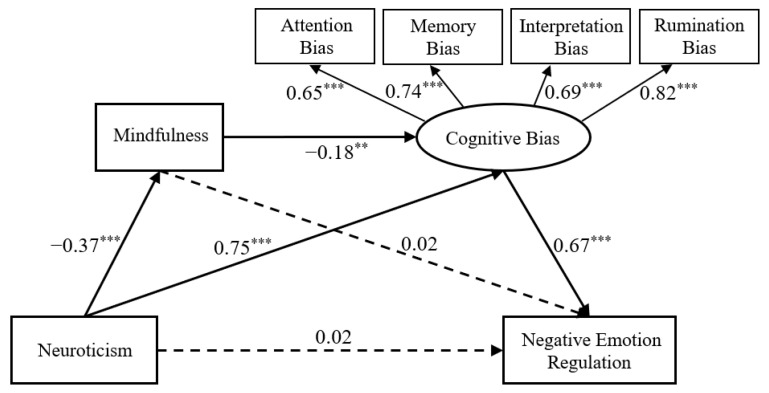
Path coefficient of the chain mediation model. Note. ** *p* < 0.01, *** *p* < 0.001.

**Figure 3 ijerph-20-00896-f003:**
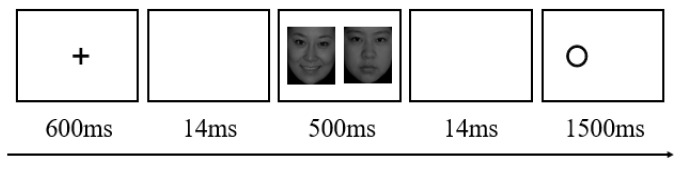
An illustrative example of the attention bias experiment.

**Figure 4 ijerph-20-00896-f004:**
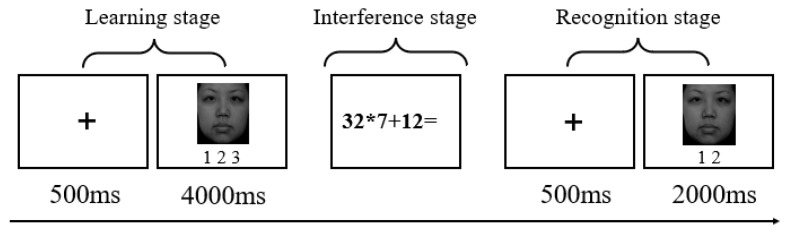
An illustrative example of the memory bias experiment.

**Figure 5 ijerph-20-00896-f005:**
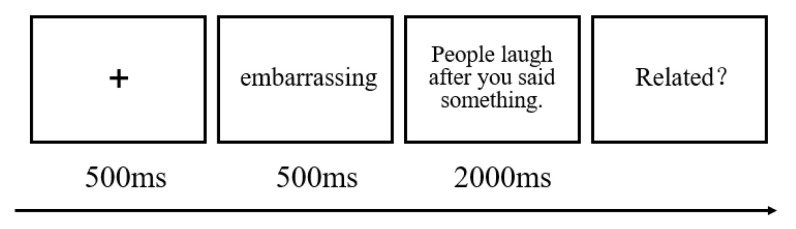
An illustrative example of the interpretation bias experiment.

**Figure 6 ijerph-20-00896-f006:**
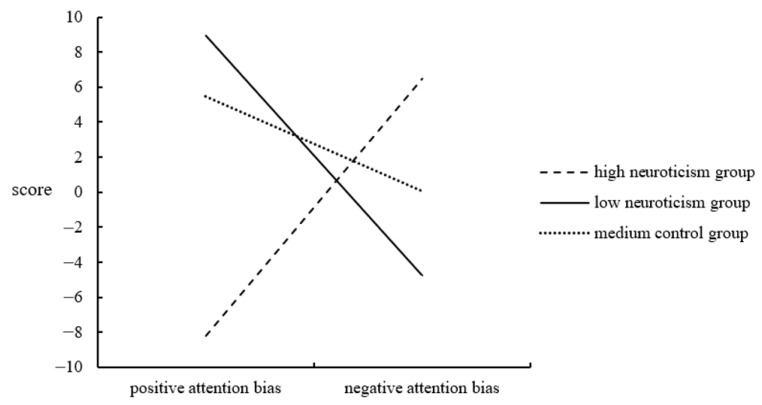
Interaction between the attention bias type and the neuroticism group.

**Figure 7 ijerph-20-00896-f007:**
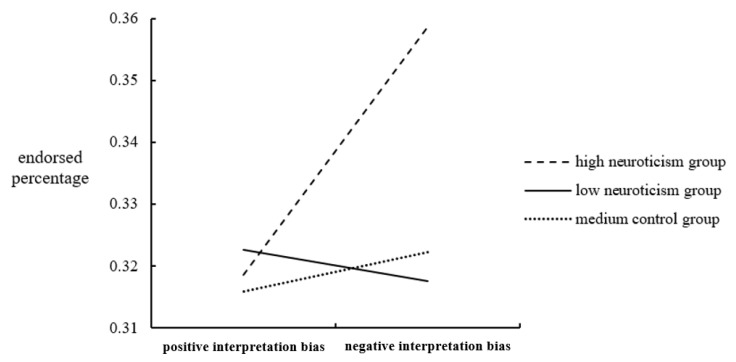
Interaction between interpretation bias type and neuroticism group.

**Table 1 ijerph-20-00896-t001:** Means, standard deviations, and correlations.

Variable	*M*	*SD*	1	2	3	4
1. Neuroticism	50.00	36.02	1			
2. Mindfulness	2.97	0.33	−0.364 ***	1		
3. Negative cognitive bias	4.32	0.74	0.734 ***	−0.388 ***	1	
4. Negative emotion regulation	3.06	0.52	0.551 ***	−0.290 ***	0.597 ***	1

Note. *** *p* < 0.001.

**Table 2 ijerph-20-00896-t002:** Bootstrap analysis of the mediation effects in the chain mediation model.

Path	Standardized Estimates	Effect Size (%)	95% CI
Total effect	0.553 ***	100	[0.445, 0.652]
Total indirect effect	0.535 ***	96.75	[0.316, 0.783]
1. Neuroticism → mindfulness → negative emotion regulation	−0.008	1.45	[−0.067, 0.036]
2. Neuroticism → negative cognitive bias → negative emotion regulation	0.500 ***	90.42	[0.283, 0.750]
3. Neuroticism → mindfulness → negative cognitive bias → negative emotion regulation	0.043 *	7.78	[0.014, 0.096]

Note. CI = confidence interval. * *p* < 0.05, *** *p* < 0.001.

**Table 3 ijerph-20-00896-t003:** Group validity test. (*M* ± *SD*; Unit: ms).

	High Neuroticism Group	Low Neuroticism Group	Medium Control Group	F	*p*
score	90.00 ± 17.89	11.33 ± 12.30	48.00 ± 4.98	281.104	0.000
N	30	30	30		

**Table 4 ijerph-20-00896-t004:** Descriptive statistics of the five experimental conditions. (*M* ± *SD*; Unit: ms).

	High Neuroticism Group	Low Neuroticism Group	Medium Control Group
positive consistent	365.52 ± 62.92	339.57 ± 38.02	370.95 ± 69.79
positive inconsistent	357.34 ± 63.05	348.48 ± 40.18	376.43 ± 83.05
negative consistent	358.04 ± 64.43	346.40 ± 41.86	372.31 ± 67.21
negative inconsistent	364.52 ± 70.97	341.64 ± 39.90	372.39 ± 58.75
neutral	357.57 ± 60.90	345.63 ± 43.28	367.73 ± 66.92

**Table 5 ijerph-20-00896-t005:** Descriptive statistics of positive and negative attention biases. (*M* ± *SD*; Unit: ms).

	High Neuroticism Group	Low Neuroticism Group	Medium Control Group
positive attention bias	−8.19 ± 13.33	8.91 ± 11.45	5.48 ± 26.18
negative attention bias	6.47 ± 16.18	−4.76 ± 10.93	0.08 ± 22.27

**Table 6 ijerph-20-00896-t006:** Descriptive statistics of the three types of emotional pictures. (*M* ± *SD*; Unit: %).

	High Neuroticism Group	Low Neuroticism Group	Medium Control Group
positive picture	20.88 ± 19.55	20.66 ± 16.55	26.66 ± 19.65
negative picture	29.78 ± 18.67	37.33 ± 21.20	32.44 ± 22.27
neutral picture	16.88 ± 14.72	20.88 ± 16.77	22.00 ± 19.43

**Table 7 ijerph-20-00896-t007:** Descriptive statistics of the four reaction time variables. (*M* ± *SD*; Unit: ms).

	High Neuroticism Group	Low Neuroticism Group	Medium Control Group
endorse threat	470.44 ± 195.72	461.59 ± 251.06	526.63 ± 275.89
reject threat	571.90 ± 336.21	510.90 ± 224.73	606.61 ± 449.04
endorse benign	524.04 ± 291.43	495.89 ± 249.09	569.41 ± 311.05
reject benign	556.50 ± 303.24	549.39 ± 311.79	556.20 ± 287.46

**Table 8 ijerph-20-00896-t008:** Descriptive statistics of positive and negative interpretation biases. (*M* ± *SD*; Unit: %).

	High Neuroticism Group	Low Neuroticism Group	Medium Control Group
positive interpretation bias	31.86 ± 4.94	32.26 ± 5.13	31.59 ± 6.22
negative interpretation bias	35.86 ± 5.41	31.76 ± 3.94	32.23 ± 6.52

## Data Availability

Data supporting this study’s findings are available from the corresponding author upon reasonable request.

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
