# Peer review of "The Emotion Regulation Mechanism in Neurotic Individuals: The Potential Role of Mindfulness and Cognitive Bias"

_ijerph, 2023, doi:10.3390/ijerph20020896_

Round 1

Reviewer 1 Report

Overall, this article is interesting and well-written. The authors define and pose questions of research well and interestingly. They choose the right methods and tools to find the answer to that questions. And above all, able to answer questions clearly and concretely.  What interests me in further discussion is mindfulness and self-direction. According to previous references, it has a very positive effect on mental health and human mechanics. You could offer suggestions that more prominently link the results of the study. that will be useful both in practice including related policy levels. 

Author Response

We would like to thank you for your insightful comments. We have put our best effort in revising the manuscript so that it provides a more rigorous methodology and a more reasonable discussion. Please find the item-by-item responses below.

Point 1: What interests me in further discussion is mindfulness and self-direction. According to previous references, it has a very positive effect on mental health and human mechanics. You could offer suggestions that more prominently link the results of the study. that will be useful both in practice including related policy levels.

Response 1: Thank you for the suggestion. We have revised Section 4 “General Discussion” accordingly. The revised text is as follows. “These results suggest that we can improve the emotion regulation of neuroticism by increasing their level of mindfulness, and then improving the cognitive bias. The reperceiving model suggests that mindfulness can regulate modes of perceiving, which allows people to focus on the moment-to-moment experience. Mindfulness reduces the distortion of the emotional valence caused by stimuli, making cognition more neutral and objective, increasing behavioral flexibility and resulting in a more positive cognitive bias. Therefore, future studies can make more efforts to investigate whether the intervention of mindfulness can reduce the negative emotion regulation of neuroticism. Furthermore, mindfulness training can be used in preventing the increase of negative emotion in neurotic individuals by improving their mental flexibility in response to the environment. The application of mindfulness training may also be generalized to non-neurotic population.”

In Section 4.3 “Limitations and future possibilities” (pp.13), we proposed some future research directions. The text is as follows. “This study has some limitations. First, it did not cover all aspects of cognitive bias in neurotic individuals. According to the neurocognitive theory of negative cognitive bias, four aspects should be included: negative attentional bias, interpretation bias, memory bias, and rumination bias. This study omitted the rumination bias. Future research could evaluate whether there are interactions among these four types of bias. Second, trait mindfulness was the only variable used in this study to quantify mindfulness. Future studies could examine the effect of mindfulness training on improving emotion regulation mechanisms. This may provide more empirical evidence for the application of cognitive behavioral therapy to neuroticism treatment. Third, this study only explored the emotion regulation of neurotic individuals through scales and behavioral experiments. Future research could integrate multiple methods such as behavioral experiments and brain imaging technology (EEG, ERP, fMRI, etc.) to investigate the emotion regulation mechanisms of neurotic individuals, such as the identification of the cortical network associated with brain mechanisms underlying emotion regulation. Fourth, the present study’s cross-sectional design and mediation analysis were unable to establish causality. Future research could design longitudinal studies to specify the directionality of the relationship between variables. Fifth, since some of the previous studies showed that people suffering from depression had higher levels of neuroticism, future studies should consider and control the role of depression. In addition, the current study only recruited university students, which might limit the generalizability of the findings. Therefore, other groups of people should be examined in future studies.”

Please see the highlighted revisions in the manuscript.

Reviewer 2 Report

This is an interesting and well-written paper, which does add to our knowledge of this topic (cognitive aspects of neuroticism in young adults), with implications for further research, and clinical application.

The authors should consider adding a final paragraph on the design of planned future studies which could flow from their research, such as longitudinal work (with students and others), and research on cognitive approaches to therapy for those who have been diagnosed with neurosis.  Since depression is part of the neurotic pattern, the role of depression might be considered, and controlled for.

The authors use an 'old' measure of neuroticism (the Eysenck scales), which is likely still a valid measure. But does the Eysenck measure correlate well with the Big 5 Neuroticism measure, referred to in their reference [54]? Apparently, they do correlate well:

McCrae, R. R., & Costa Jr, P. T. (1985). Comparison of EPI and psychoticism scales with measures of the five-factor model of personality. Personality and individual Differences6(5), 587-597.

Why didn't the authors of the present study use the five-factor measures of personality at the outset? This might be explained.

The title of the study under review might be changed to: "The Potential Role of  ..." rather than "Effects of ...". As the authors correctly point out in the Limitations section, it is difficult to be sure about causal pathways in a cross-sectional study.

Author Response

We would like to thank you for your insightful comments. We have put our best effort in revising the manuscript so that it provides a more rigorous methodology, and a more reasonable discussion. Please find the item-by-item responses below.

Point 1: The authors should consider adding a final paragraph on the design of planned future studies which could flow from their research, such as longitudinal work (with students and others), and research on cognitive approaches to therapy for those who have been diagnosed with neurosis.  Since depression is part of the neurotic pattern, the role of depression might be considered, and controlled for.

Response 1: Thank you for the suggestion. We have enriched Section 4.3 ”Limitations and future possibilities” (pp.13) and proposed future research directions. The text is as follows. “This study has some limitations. First, it did not cover all aspects of cognitive bias in neurotic individuals. According to the neurocognitive theory of negative cognitive bias, four aspects should be included: negative attentional bias, interpretation bias, memory bias, and rumination bias. This study omitted the final component (rumination bias). Future research could evaluate whether there are interactions among these four types of bias. Second, trait mindfulness was the only variable used in this study to quantify mindfulness. Future studies could examine the effect of mindfulness training on improving emotion regulation mechanisms. This may provide more empirical evidence for the application of cognitive behavioral therapy to neuroticism treatment. Third, this study only explored the emotion regulation of neurotic individuals through scales and behavioral experiments. Future research could integrate multiple methods such as behavioral experiments and brain imaging technology (EEG, ERP, fMRI, etc.) to investigate the emotion regulation mechanisms of neurotic individuals, such as the identification of the cortical network associated with brain mechanisms underlying emotion regulation. Fourth, the present study’s cross-sectional design and mediation analysis were unable to establish causality. Future research could design longitudinal studies to specify the directionality of the relationship between variables. Fifth, since some of the previous studies showed that people suffering from depression had higher levels of neuroticism, future studies should consider and control the role of depression. In addition, the current study only recruited university students, which might limit the generalizability of the findings. Therefore, other groups of people should be examined in future studies.” Please see the highlighted revisions in the manuscript.

Point 2: The authors use an 'old' measure of neuroticism (the Eysenck scales), which is likely still a valid measure. But does the Eysenck measure correlate well with the Big 5 Neuroticism measure, referred to in their reference [54]? Apparently, they do correlate well:

McCrae, R. R., & Costa Jr, P. T. (1985). Comparison of EPI and psychoticism scales with measures of the five-factor model of personality. Personality and individual Differences, 6(5), 587-597.

Why didn't the authors of the present study use the five-factor measures of personality at the outset? This might be explained.

Response 2: Thank you for pointing out this aspect of the manuscript. We have referred to relevant literature and compared the Eysenck Personality Questionnaire (EPQ) and Big Five Inventory (BFI) carefully. First, we found that both scales are widely used and suitable for measuring the neuroticism of undergraduates in China. Second, each scale has a brief version, which was tested and showed great reliability and validity.

There are still some differences between the two scales, which resulted in our choice of EPQ. First of all, EPQ was initially developed by Eysenck just for measuring neuroticism, which was exactly what we focused on. One of the most solid theoretical and empirical models of personality was developed by Eysenck [1-3], who described personality based on three independent dimensions: Neuroticism, Extraversion and Psychoticism. Second, EPQ has been used in measuring neuroticism to date. Some recent studies [4-6] and those cited by our article [7-9] used EPQ to measure neuroticism. What’s more, our laboratory has the copyright of EPQ instead of BFI, which makes it more convenient to implement the measure.

However, we fully agree that we should consider the timeliness of the scales. We will think much more seriously before we choose the scale for measurement next time.

Point 3: The title of the study under review might be changed to: "The Potential Role of  ..." rather than "Effects of ...". As the authors correctly point out in the Limitations section, it is difficult to be sure about causal pathways in a cross-sectional study.

Response 3: Thank you for your suggestion. We have rephrased the title to: “The Emotion Regulation Mechanism in Neurotic Individuals: The Potential Role of Mindfulness and Cognitive Bias”.

References

  1. Eysenck, H. J. The biological basis of personality; Springfield: Massachusett, USA, 1967.
  2. Eysenck, H. J. The structure of human personality (3 ed.); Methuen: London, UK, 1970.
  3. Eysenck, H. J. A model for personality; Springer-Verlag: Berlin, Ger, 1981.
  4. Sauer, S.; Fournier, J.; Steele, S. J.; Woods, B.; David, B. Does the unified protocol really treat neuroticism? Results from a randomized-controlled trial. Psychol. Med. 2021, 51(14), 2378-2387, doi: 10.1017/S0033291720000975.
  5. Reid, M. J.; Omlin, X.; Espie, C. A.; Sharman, R.; Tamm, S. The effect of sleep continuity disruption on multimodal emotion processing and regulation: A laboratory-based, randomized, controlled experiment in good sleepers. J. Sleep Res. 2022, e13634, doi: 10.1101/2022.04.22.489209.
  6. Murakami, K.; Ishikuro, M.; Obara, T.; Ueno, F.; Noda, A.; Onuma, T.; Matsuzaki, F.; Kikuchi, S.; Kobayashi, N.; Hamada, H.; Iwama, N.; Metoki, H.; Saito, M.; Sugawara, J.; Tomita, H.; Yaegashi, N.; Kuriyama, S. Maternal personality and postpartum mental disorders in Japan: The Tohoku Medical Megabank Project Birth and Three-Generation Cohort Study. Sci. Rep. 2022, 12(1), 6400:1-6400:8, doi: 10.1038/s41598-022-09944-w.
  7. Denovan, A.; Dagnall, N.; Lofthouse, G. Neuroticism and somatic complaints: Concomitant effects of rumination and worry. Behav. Cogn. Psychother. 2019, 47(4), 431-445, doi: 10.1017/S1352465818000619.
  8. Rijsdijk, F.V.; Riese, H.; Tops, M.; Snieder, H.; Brouwer, W.H.; Smid, H.G.O.M.; Ormel, J. Neuroticism, recall bias and attention bias for valenced probes: A twin study. Psychol. Med. 2009, 39, 45-54, doi: 10.1017/S0033291708003231.
  9. Armstrong, L.; Rimes, K. A. Mindfulness-based cognitive therapy for neuroticism (stress vulnerability): A pilot randomized study. Behav. Ther. 2016, 47(3), 287-298, doi: 10.1016/j.beth.2015.12.005.
